# Is It Possible to Improve Urinary Incontinence and Quality of Life in Female Patients? A Clinical Evaluation of the Efficacy of Top Flat Magnetic Stimulation Technology

**DOI:** 10.3390/bioengineering9040140

**Published:** 2022-03-25

**Authors:** Graziella Lopopolo, Benedetta Salsi, Alessandra Banfi, Pablo González Isaza, Irene Fusco

**Affiliations:** 1Division of Gynecology, Poliambulatorio San Michele, 42121 Reggio Emilia, Italy; graziella.lopopolo@gmail.com (G.L.); allebanfi@yahoo.it (A.B.); 2Division of Dermatology, Poliambulatorio San Michele, 42121 Reggio Emilia, Italy; slsbdt@gmail.com; 3Functional and Regenerative Gynecology DIATROS, Clínica d’ Atenció a la Dona, 08022 Barcelona, Spain; pagonza@hotmail.com; 4Department of Pharmacology, University of Florence, 50121 Florence, Italy

**Keywords:** mixed urinary incontinence, Top Flat Magnetic Stimulation, hypotonus

## Abstract

*Background and Objectives*: Urinary incontinence is the accidental loss of urine that can occur at any age, especially among women over 50; however, its prevalence is increasing. This study aimed to assess the efficacy and safety of a device that uses Top Flat Magnetic Stimulation for the management of women with mixed urinary incontinence. *Materials and Methods*: A total of 50 female patients underwent a total of six treatment sessions performed twice a week for three consecutive weeks, for 28 min. Three questionnaires were used for the evaluation of the urinary improvements: Urinary Incontinence Short Form (ICIQ-UI-SF), Incontinence Questionnaire Overactive Bladder Module (ICIQ-OAB), and Incontinence Impact Questionnaire-Short Form (IIQ-7). The questionnaires were compiled immediately before each treatment, within the treatment period (until the sixth treatment session), and up to 3 months of follow-up. *Results:* Improvement in quality of life is observed according to the results of the questionnaire, already after six sessions and at three months follow-up. ICIQ-UI-SF’s average score at baseline was 20.2 (±1.1), and it significantly declined (*p* < 0.01) to 1.8 (±2.4) after six sessions and to 4.8 (±2.8) at 3 months follow-up; ICIQ-OAB’s average score significantly decreased (*p* < 0.01) from 10.4 (±3.2) at baseline to 1.4 (±0.8) after six sessions and 2.3 (±1.6) at 3 months follow-up. IIQ-7’s average score significantly decreased (*p* < 0.01) from 20.2 (±1.7) at baseline to 0.4 (±0.5) after six sessions and to 4.7 (±2.8) at 3 months follow-up. *Conclusions*: Our results suggest that Top Flat Magnetic Stimulation technology can reduce mixed urinary incontinence symptoms for all women examined, leading to an improvement in the patient quality of life and patient awareness of their pelvic floor area with good results.

## 1. Introduction

Urinary incontinence (UI) is an observation of involuntary loss of urine on examination [1].

UI is not an inevitable result of aging, but it is widespread in the adult age group. It is secondary to body function modifications that may result from diseases, medications, and illness. It may be the only symptom of a urinary tract infection. Women are most likely to develop UI during pregnancy, after childbirth, or after the hormonal changes of menopause [2].

The pelvic floor is affected by complex changes due to pregnancy, postpartum, and aging. Pelvic floor muscles (PFMs) do not provide enough support for pelvic organs and continence mechanisms. This leads to PFM dysfunction directly resulting in UI. Involuntary urine leakage may interest all ages of life, particularly older women living in nursing homes. Some females may leak urine during exercise or when they cough or sneeze (stress urinary incontinence, SUI). This may occur due to weakness of the pelvic floor muscles due to other factors, such as childbirth associated damage. Other females may experience this problem before going to the bathroom when there is a sudden and compelling need to urinate (urgency urinary incontinence, UUI). This may be also associated with involuntary contraction of the bladder muscle. Mixed urinary incontinence (MUI) combines stress and urgency urinary incontinence [3].

Unfortunately, there is a wide range of functional problems related to pelvic floor dysfunction, which concern, in addition to UI (10%–52%), anorectal disorders (fecal incontinence and obstructed defecation) (10%–20%), genital prolapse (10%–20%), urgency, urinary fecal, pain, and sexual dysfunctions [4,5].

Historically, the diagnosis and treatment of pelvic floor disorders was divided “into sectors” with precedence to the predominant disorder and treated by both urologists and gynecologists. Only in complex cases, were coloproctologists also consulted. Not all cases of UI require a complete pelvic floor evaluation. However, from the outset, it may be helpful to identify the cases that are susceptible to this assessment. Schematically, the diagnostic approach to female UI can be divided into an initial, purely clinical evaluation and an advanced evaluation involving specific instruments.

UI treatment strategies include surgical treatments, conservative/physical therapies (vaginal cones or pessaries, timed voiding, and fluid restriction), pharmacological approach (anticholinergic medications, vaginal estrogens, botulin toxin injections into bladder, urethral bulking agents, and peripheral nerve stimulation), and behavioral therapies, such as pelvic muscle rehabilitation including Kegel exercises, biofeedback, vaginal weight training, and pelvic floor electrical stimulation [6]. Surgical treatment or application of urethral sling/periurethral injection of bulking agents [7] improves incontinence in approximately 90% of patients; however, this treatment modality is invasive and may involve risks and complications [8], leading patients to be naturally reluctant to consider it. Techniques, such as pelvic floor muscle training, electrical stimulation, and biofeedback, are effective for 60% of individuals with SUI. Although some benefits of physiotherapy are known, its disadvantages are represented by a slow progression, low compliance, and patient adherence rates [9]. Indeed, the effectiveness of Kegel exercises is reduced because they are often not performed correctly and constantly over time by the patients (women often need to be motivated to perform Kegel exercises routinely) [10].

In recent years, magnetic stimulation has been investigated as an alternative treatment to electrical stimulation (ES) for clinical neurodiagnostic applications and urological diseases as a safe and noninvasive method for nerve tissue stimulation [11,12,13].

Among the advantages of magnetic stimulation, in comparison to ES is the fact that for the same current generated at the nerve root level, magnetic stimulation requires less current to be generated at the body surface. Therefore, magnetic stimulation can activate deep neural structures by inducing electric currents without patient discomfort or pain. Moreover, the deep penetration of the electromagnetic field into the pelvic area results in greater pelvic floor musculature strength (PFMS) activation than electrostimulators that lose the most significant portion of emitted energy on the surface, and only its fraction reaches deep-lying tissues.

Furthermore, several studies demonstrate that magnetic stimulation has been shown to be effective in treating SUI/UUI conditions [11,14,15,16,17,18] and in MUI patients [19,20,21].

In addition, the need to stimulate the entire pelvic floor evenly without having areas of greater stimulation intensity than others led to the creation of systems with more flat and extended emissions.

On these bases, this study shed light on the evaluation of the efficacy and safety of a device that uses Top Flat Magnetic Stimulation for the management of women with MUI.

## 2. Materials and Methods

### 2.1. Patient Population

Fifty female patients who met the criteria for MUI with an average age of 65.33 ± 20.42 years were enrolled for this study. Participants were classified as MUI patients with the relevant questions present in the questionnaires selected for the study regarding inclusive criteria for three types of UI, which were designed according to the UI classification of the International Continence Society [22].

The demographics of the study sample are presented in Table 1.

All patients were informed about the treatment procedure, the number of sessions needed to achieve the expected results, the steps to follow before each treatment session, and any possible side effects resulting from treatment. Inclusion criteria: Participants will be enrolled if they are: ≥40 years of age; educated enough to complete the questionnaires; previously diagnosed as MUI.

Exclusion criteria included: patients with cardiac pacemakers, implanted defibrillators/neurostimulators, electronic/metal implants, pulmonary insufficiency, heart disorders, severe neurological diseases, malignant tumor, urinary or genital tract infections, pregnancy, and obesity.

### 2.2. Dr. ARNOLD System and Patient Position

Dr. ARNOLD (DEKA M.E.L.A. Calenzano, Italy) is a noninvasive therapeutic system that selectively stimulates the female pelvic floor muscles through programmed specific contractions and relaxations to improve UI. The stimulation is generated by electromagnetic fields with a homogenous profile (TOP FMS–TOP Flat Magnetic Stimulation) optimized for stimulation of the pelvic area. The interaction with the tissue includes muscular contraction, depolarization of neuronal cells, and influence on the blood circulatory system.

The Dr. ARNOLD system consists of a main unit and a chair applicator designed for deep pelvic floor area therapy.

The Dr. ARNOLD system has been CE marked since July 2020, and it is a medical device indicated for pelvic floor muscle strengthening and urinary incontinence. The coil is located in the center of the seat. The patient is positioned on the chair to locate the patient’s perineum on the seat center; this allows the patient to feel muscle contraction (both of the pelvic floor and sphincter muscles) during treatment stimulation.

In the early stage of treatment, the patient’s position was set and adjusted before each session by a gynecologist to ensure an adequate stimulation intensity and to reach a homogeneous distribution of muscle contractions.

The change of magnetic fields, generated by the device, transmits current directly to the muscle tissue in depth, contracting it and releasing it. An advantage of the device is due to the greater homogeneity of magnetic field distribution in a broader area, which allows greater recruitment of muscle fibers without creating areas of variable stimulation intensity. Thanks to this technology, the muscle works at the same intensity in all the fields to be treated. The physician sets both the stimulation intensity and the patient chair position before each treatment to guarantee adequate stimulation. The correct patient position on the chair represents a crucial aspect, both to ensure patient comfort during the entire duration of the treatment and to center the electromagnetic field on the patient’s perineum efficaciously. For precise patient position prior to the therapy, the seat height was adjusted, setting the height where the patient’s legs are perpendicularly flexed; thighs are parallel to the floor, and feet are flat on the ground. Patients should form an angle of 90 degrees or slightly higher at the knee (Figure 1).

### 2.3. Study Protocol and Assessments

All patients underwent 6 treatments with Dr. ARNOLD system (DEKA M.E.L.A. Calenzano, Italy). Sessions were performed twice a week for 3 consecutive weeks, for 28 min. During the first two minutes, all patients underwent a warm-up phase; the two protocols, Hypotonus/Weakness 1 (muscle work aimed to recover trophism and muscle tone) and Hypotonus/Weakness 2 (muscle work aimed to increase trophism (volume) and muscle strength) were used.

To ensure adequate stimulation, the doctor set the position of the patient chair before each treatment and the intensity of the stimulation.

The status and severity of each patient’s mixed urinary incontinence was assessed by a gynecologist.

Three questionnaires were used for the evaluation of the urinary improvements: Urinary Incontinence Short Form (ICIQ-UI-SF) [23], Incontinence Questionnaire Overactive Bladder Module (ICIQ-OAB) [24], and Incontinence Impact Questionnaire-Short Form (IIQ-7) [25] (see Table 2). The questionnaires were completed immediately before each treatment, within the treatment period (until the sixth treatment session) and at three months follow up (FU).

The results of the questionnaires were calculated and statistically evaluated. Generally, a high score indicates an important presence of symptoms and a significant impact of incontinence on daily life, so the decrease indicates improvement.

The means of the scores of the 3 questionnaires were calculated at each session, until the 6th treatment session and at 3 months FU.

The onset of possible side effects, such as muscular pain, temporary joint or tendon pain, temporary muscle spasm, and local erythema or skin redness, were evaluated during all treatment periods.

### 2.4. Statistical Analysis

Student’s *t*-test and SPSS (IBM Corp., New York, USA) were used to perform statistical analysis. Data were represented as means ± standard deviation (SD).

## 3. Results

Improvement in quality of life is observed, according to questionnaire results, at baseline, after each treatment session (until the 6th treatment session) and at 3 months follow-up, as shown in Table 3 where questionnaire mean scores (±SD) with relative percentage variations were reported. All three questionnaire mean scores significantly decreased (*p* < 0.01) from the second treatment session.

ICIQ-UI-SF’s average score at baseline was 20.2 (±1.1), and it significantly declined (*p* < 0.01) to 1.8 (±2.4) (with a percentage change of 91%) after six sessions (Figure 2) and to 4.8 (±2.8) at 3 months follow-up (Figure 3)

At the baseline evaluation, patients most frequently reported experiencing leakage several times a day. After six sessions, the leakage occurred only about once a week or less. There is also a variation in the level of severity of the disease, which varies from very severe (at baseline) to slight after six sessions; ICIQ-OAB’s average score significantly decreased (*p* < 0.01) from 10.4 (±3.2) at baseline to 1.4 (±0.8)(with a percentage change of 86%) after six sessions (Figure 4) and to 2.3 (±1.6) at 3 months follow-up (Figure 5); there was an improvement in the nocturia disorder and better management of urinary frequency and urgency; lastly, IIQ-7′s average score significantly decreased (*p* < 0.01) from 20.2 (±1.7) at baseline to 0.4 (±0.5) (with a percentage change of 98%) after six sessions (Figure 6) and to 4.7 (±2.8) at 3 months follow-up (Figure 7).

According to the answers, at the baseline, the patients’ health condition and relationships were affected, and social and emotional limitations were present. No side effects were observed.

## 4. Discussion

The results of this study clearly show the benefits of Top Flat Magnetic Stimulation for MUI. Our data indicate an improvement after the first treatment, which increases and persists until the end of the study. Already from the second treatment session, all questionnaire means scores significantly decreased, and a percentage variation of 91% for ICIQ-UI-SF scores, 86% for ICIQ-OAB, and 98% for IIQ-7 was observed in all patients examined after six treatment sessions.

Subjects benefited from a significantly decreased severity of MUI symptoms and a reduced usage of sanitary pads, which positively influenced their quality of life. Based on the subjective evaluation, patients also reported additional effects of the therapy, such as better control of urination.

On the bases of our findings, the device might be defined as an “educator” system as it allows for greater patient perception about the contraction and relaxation of the muscles involved during the treatment and a greater patient awareness and autonomy to repeat the following treatment session.

At 3 months follow-up, a slight increase in scores was observed, although with optimal results compared to the baseline; this presumably was due to the absence of long-term exercise that leads to a physiological hypotonus of the patient.

Several UI treatments, such as surgery, may not be the optimal therapeutic choices for all patients; surgical procedures are more likely to be curative [26], but are more invasive and have more associated adverse events; furthermore, surgery is often associated with multiple contraindications, including anesthesia risks, use of anticoagulants, urinary infection, as well as a patient’s general apprehension and worry regarding the risks associated with these procedures.

This therapeutic approach has spawned the research and development of minimally invasive energy-based devices that could be used as an alternative treatment option to more effectively address UI in females. Particularly, magnetic stimulation technology has been well established in this field as reported in several studies [27,28,29,30], which proved that it is a noninvasive UI treatment option, leading to a significant reduction in UI frequency and showing short- and long-term improvements in overall and various physical, social, and psychological aspects of QoL, particularly in physical activities. A recently published review shows the effectiveness of MS technology on all types of UI, including MUI, with similar assessments and results [31].

Instead of feeling burdened with social stigmatization and embarrassment, an improved QoL means these women can live a more active lifestyle with reduced physical and emotional stress.

This novel technology triggers intense pelvic floor muscle (PFM) contractions by targeting neuromuscular tissue and inducing electric currents. Electric currents depolarize neurons resulting in concentric contractions and lifting up all PFMs. Key effectiveness is based on electromagnetic energy, in-depth penetration, and stimulation of the entire pelvic floor area. This directly changes the muscle structure, inducing more efficient growth of myofibrils (muscle fiber hypertrophy) and creating new protein filaments and muscle fibers (muscle fiber hyperplasia). Magnetic stimulator technology causes deep PFM stimulation and restoration of neuromuscular control. Repeated activation of the terminal motor nerve fibers and the motor end plates will tend to build muscle strength and endurance [32]. MS creates a rapidly pulsating magnetic field whose frequency and pulsation strength can be adjusted on the device. The roots of sacral nerves S2–S4 provide the primary autonomic and somatic innervation of the urinary bladder and urethra, vaginal wall and rectum, and pelvic floor muscles. Stimulation of these roots is an efficient way to modulate the pelvic floor and subsequently control the pelvic organs [33]. This method is used for treating all types of urinary incontinence.

Compared to different devices for pelvic floor restoration, the subject device introduces some essential advantages. First of all, it represents a noninvasive technology, as it does not introduce any probe in the vaginal canal for muscle stimulation. Therefore, thanks to the regular emission of energy progressively delivered, patients can stay fully clothed in a comfortable and ergonomic seat, and the uninterrupted treatments allow patients to resume their daily activities immediately after the sessions.

The other two key aspects are represented by the patient’s position (the chair is equipped with a height-adjustable backrest in order to allow the patient to reach the optimal position to benefit from the treatment, in complete comfort and relaxation) and the possibility to have a gradual increase in the intensity of electromagnetic fields and of pulses frequency, which determine a unique vigorousness of muscle contractions. Indeed, the treatment effectiveness is linked to the ability to contract the affected muscles correctly.

This device can also be used in combination with other physical or pharmacological methods [34].

### Study Limitation

Limitations of the current study include the lack of a long-term follow-up. Further studies with a longer follow-up are needed to better confirm our findings. Therefore, it may be useful to increase the sample size and follow-up period to evaluate the long-term efficacy of the method.

Finally, we examined patients with MUI for a first evaluation; the next step will be to identify patients only affected by SUI or UUI condition.

## 5. Conclusions

Our findings suggest that Top Flat Magnetic Stimulation technology could represent a novel treatment option for MUI. These treatment protocols, which proved to be minimally invasive and without risk, led to a reduction in MUI symptoms in all women examined, achieving good results, and therefore, an improvement in patient quality of life and patient awareness of the pelvic floor area.

## Figures and Tables

**Figure 1 bioengineering-09-00140-f001:**
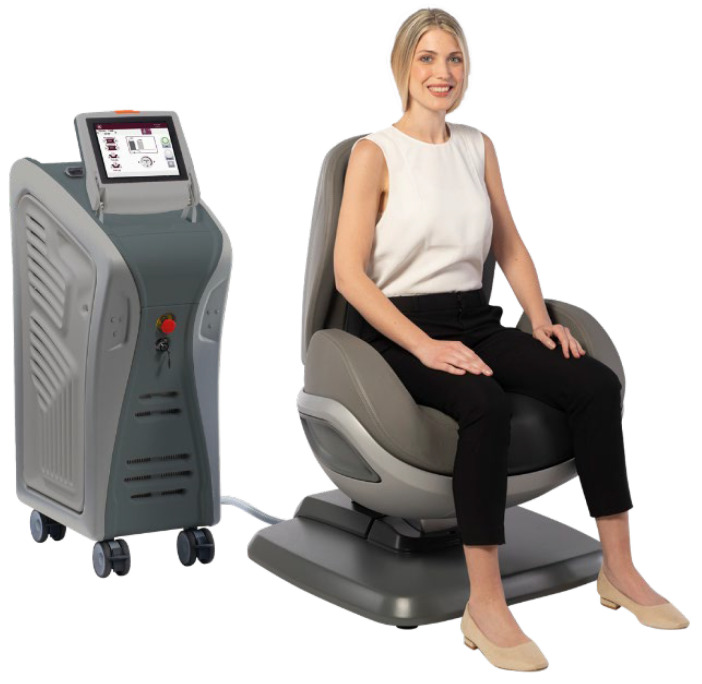
Representation of the correct patient position on the Dr. ARNOLD chair. Courtesy of DEKA M.E.L.A company.

**Figure 2 bioengineering-09-00140-f002:**
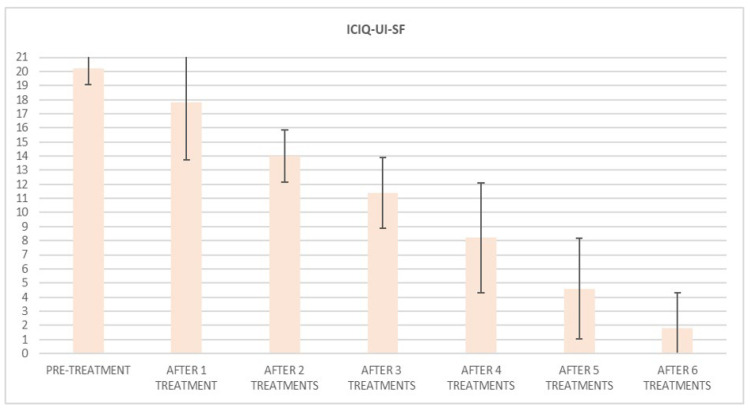
Histogram representation of results from ICIQ-UI-SF at baseline and at each treatment session until the 6th treatment session.

**Figure 3 bioengineering-09-00140-f003:**
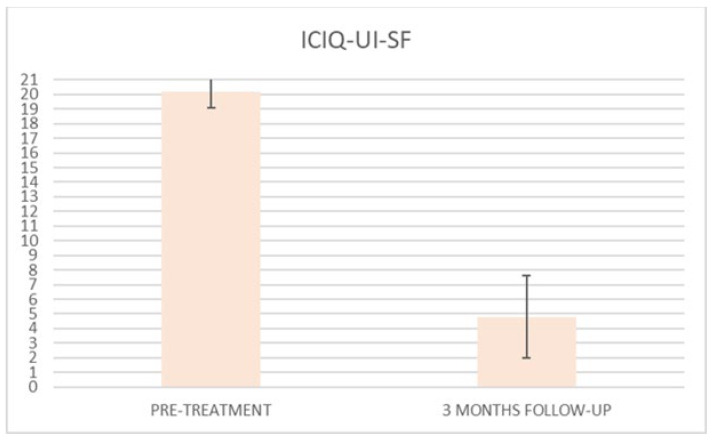
Histogram representation of the results from ICIQ-UI-SF at baseline and at 3 months follow-up.

**Figure 4 bioengineering-09-00140-f004:**
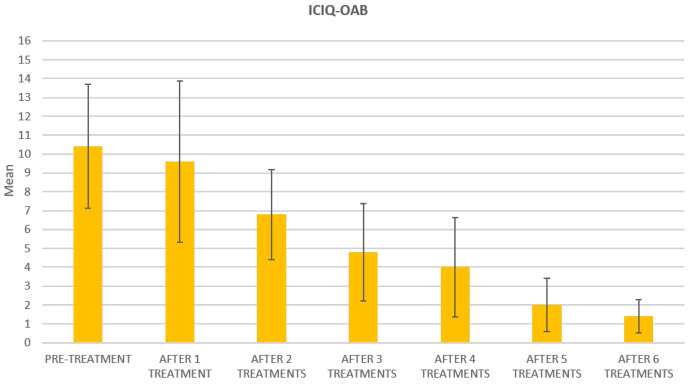
Histogram representation of the results from ICIQ-OAB at baseline and at each treatment session until the 6th treatment session.

**Figure 5 bioengineering-09-00140-f005:**
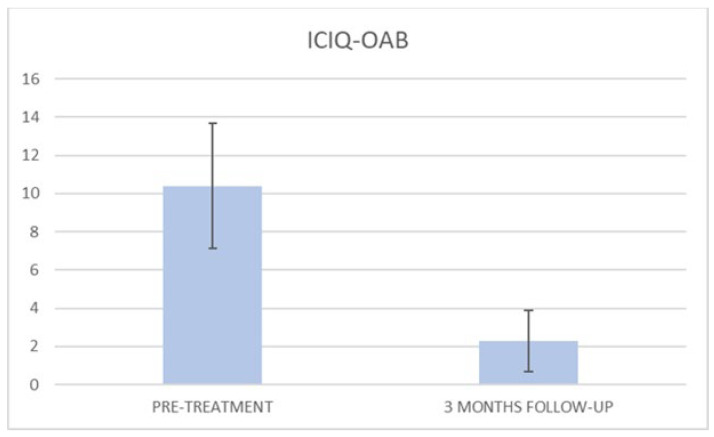
Histogram representation of the results from ICIQ-OAB at baseline and at 3 months follow-up.

**Figure 6 bioengineering-09-00140-f006:**
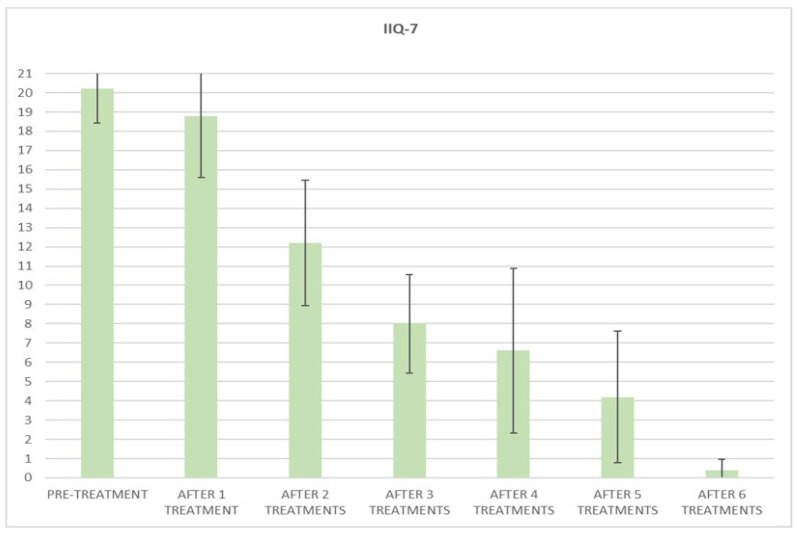
Histogram representation of the results from IIQ-7 at baseline and at each treatment session until the 6th treatment session.

**Figure 7 bioengineering-09-00140-f007:**
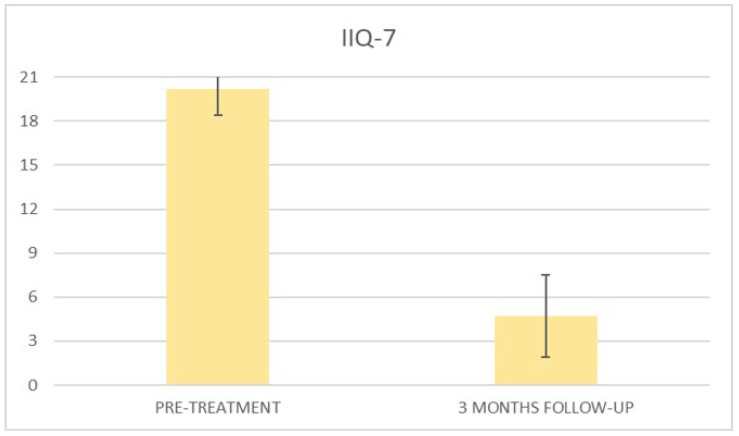
Histogram representation of the results from IIQ-7 at baseline and at 3 months follow-up.

**Table 1 bioengineering-09-00140-t001:** Demographic Characteristics of patients.

**N° of Patients**	50
**UI Type**	MUI
**Menopausal Patients**	30
**Average age** **(Mean ± SD)**	65.33 ± 20.42
**Duration of symptoms** **years** **(Mean ± SD)**	4.70 ± 1.30
**BMI** **kg/m^2^**	24.60

**Table 2 bioengineering-09-00140-t002:** Evaluation questionnaires.

	Score Range	Aim
**ICIQ-UI-SF**	0–21, Divided into the following four severity categories: -Slight (1–5) -Moderate (6–12) -Severe (13–18) -Very severe (19–21).	Evaluation of clinical manifestations of urinary incontinence, severity of urinary loss, and impact on quality of life
**ICIQ-OAB**	0–16	For overactive bladder, evaluation of urgency, frequency, nocturia, and urgency leakage
**IIQ-7**	0–21	Evaluation of impact of urinary incontinence on activities, relationships, and emotional states

**Table 3 bioengineering-09-00140-t003:** Mean score related to ICIQ-UI-SF, ICIQ-OAB, and IIQ-7, at baseline, after each treatment session (until the 6th treatment session), and at 3 months follow-up.

	ICIQ-UI-SF	ICIQ-OAB	IIQ-7
**Baseline** **Mean ± SD**	20.2 ± 1.1	10.4 ± 3.2	20.2 ± 1.7
**After 1 treatment** **Mean ± SD**	17.8 ± 4.0	9.6 ± 4.2	18.8 ± 3.1
**After 2 treatments** **Mean ± SD**	14.0 ± 1.8 *	6.8 ± 2.3 *	12.2 ± 3.2 *
**After 3 treatments** **Mean ± SD**	11.4 ± 2.5 *	4.8 ± 2.5 *	8 ± 2.5 *
**After 4 treatments** **Mean ± SD**	8.2 ± 3.8 *	4.0 ± 2.6 *	6.6 ± 4.2 *
**After 5 treatments** **Mean ± SD**	4.6 ± 3.5 *	2.0 ± 1.4 *	4.2 ± 3.4 *
**After 6 treatments** **Mean ± SD**	1.8 ± 2.4 *	1.4 ± 0.8 *	0.4 ± 0.5 *
**Percentage change** **between baseline and up to 6 treatments score**	91%	86%	98%
**3 Months FU** **Mean ± SD**	4.8 ± 2.8 *	2.3 ± 1.6 *	4.7 ± 2.8 *

* *p* < 0.01 vs baseline.

## Data Availability

Data that support the study findings are available on request from the corresponding author.

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
