# Peer review of "Is It Possible to Improve Urinary Incontinence and Quality of Life in Female Patients? A Clinical Evaluation of the Efficacy of Top Flat Magnetic Stimulation Technology"

_bioengineering, 2022, doi:10.3390/bioengineering9040140_

Round 1

Reviewer 1 Report

The current study aims to the efficacy of Top Flat Magnetic Stimulation technology to improve urinary incontinence and Quality of Life in Female Patients.

there are some concerns that may be resolved:

Introduction:

  • Please use the ICS glossary for UI definition, and cite the appropriate reference.
  • The authors should provide a literature review to state the gaps between the findings and the necessity to do this research.
  • Method:
  • what was the study design? Please state it. 
  • How did you select the women to include in the study? what were the criteria for diagnosis of the MUI?
  • How did you calculate the sample size?
  • The inclusion criteria should be completely provided. did you consider any age category, gravida, parity, 
  • How did you control the confounders? 
  • Did you receive an ethical code to conduct this study?
  • How did you control the normal distribution of data? and in the cases with non-normal distribution, you use which statistical method?
  • Results:
  • The baseline and demographic characteristics of patients are absent. Please provide it. 
  •  What was the previous treatment method for the women?
  • The duration of symptoms should be stated. 
  • The amount of p-value should be mentioned in the Tables. 
  • Discussion:
  • Please provide the main findings of your study in the first paragraph of the discussion. 
  • The literature review should be completed and the results of the study should be compared with previously published papers.
  • The mechanism of the effect of Magnetic Stimulation on MUI should be clearly stated.
  • The neural control of urination is better to be described. 
  • What were the strengths of your study?
  • What is the recommendation for future studies?

Reviewer 2 Report

It was an interesting study about the clinical evaluation of application of a top flat magnetic stimulation technology for improving the urinary incontinence of females. Here are some comments on this study that should be considered before publication:

  1. The selected keywords are not good. You didn't introduce MUI before. Please use other keywords.
  2. There are some grammatical mistakes in the text that should be corrected.
  3. In your opinion, is it needed to repeat the treatment process after some months to prevent recurrence?
  4. Did you also check the probable effects (especially side-effects) of the treatment on body organs, especially in the targeted site?

Round 2

Reviewer 2 Report

Thanks for addressing comments.